# Extraction of radiographic findings from unstructured thoracoabdominal computed tomography reports using convolutional neural network based natural language processing

**Mohit Pandey**[1¤a], **Zhuoran Xu**[1], **Evan Sholle**[2], **Gabriel Maliakal**[1¤b], **Gurpreet Singh**[1¤c], **Zahra Fatima**[1], **Daria Larine**[3], **Benjamin C. Lee**[1], **Jing Wang**[1], **Alexander R. van Rosendael**[1¤d], **Lohendran Baskaran**[1¤e], **Leslee J. Shaw**[1], **James K. Min**[1¤b], **Subhi J. Al'Aref**[4]*

1 Department of Radiology, Weill Cornell Medicine, New York, New York, United States of America, 2 Information Technologies and Services, Weill Cornell Medicine, New York, New York, United States of America, 3 Jaffe Food Allergy Institute, Icahn School of Medicine at Mount Sinai, New York, New York, United States of America, 4 Division of Cardiology, Department of Medicine, University of Arkansas for Medical Sciences, Little Rock, Arkansas, United States of America

¤a Current address: Ipsos US Public Affairs, New York, New York, United States of America
¤b Current address: Cleerly, Inc, New York, New York, United States of America
¤c Current address: GlaxoSmithKline, Pennsylvania, Pennsylvania, United States of America
¤d Current address: Department of Cardiology, Leiden University Medical Center, Leiden, The Netherlands
¤e Current address: Department of Cardiovascular Medicine, National Heart Centre Singapore, Singapore, Singapore
* SJAlaref@UAMS.edu

**Data Availability Statement:** All relevant data are within the manuscript and its Supporting Information files.

## Abstract

### Background

Heart failure (HF) is a major cause of morbidity and mortality. However, much of the clinical data is unstructured in the form of radiology reports, while the process of data collection and curation is arduous and time-consuming.

### Purpose

We utilized a machine learning (ML)-based natural language processing (NLP) approach to extract clinical terms from unstructured radiology reports. Additionally, we investigate the prognostic value of the extracted data in predicting all-cause mortality (ACM) in HF patients.

### Materials and methods

This observational cohort study utilized 122,025 thoracoabdominal computed tomography (CT) reports from 11,808 HF patients obtained between 2008 and 2018. 1,560 CT reports were manually annotated for the presence or absence of 14 radiographic findings, in addition to age and gender. Thereafter, a Convolutional Neural Network (CNN) was trained, validated and tested to determine the presence or absence of these features. Further, the

**Funding:** This study received support from the New York-Presbyterian Hospital (NYP) and Weill Cornell Medicine (WCM), including the Clinical and Translational Science Center (CTSC) (UL1 TR000457) and Joint Clinical Trials Office (JCTO). The funders had no role in study design, data collection and analysis, decision to publish, or preparation of the manuscript. The funder provided support in the form of salaries for author [E.S.] but did not have any additional role in the study design, data collection and analysis, decision to publish, or preparation of the manuscript. The specific roles of these authors are articulated in the 'author contributions' section.

**Competing interests:** The authors have declared that no competing interests exist. Gurpreet Singh is currently employed at GlaxoSmithKline but was not a part of GlaxoSmithKline during the conduct of this study. Gabriel Maliakal and James K. Min are currently employed at Cleerly Inc. but were not a part of Cleerly Inc. during the conduct of this study. Mohit Pandey is currently employed at Ipsos but was not a part of Ipsos during the conduct of this study. These commercial affiliations do not alter our adherence to PLOS ONE policies on sharing data and materials.

ability of CNN to predict ACM was evaluated using Cox regression analysis on the extracted features.

## Results

11,808 CT reports were analyzed from 11,808 patients (mean age 72.8 ± 14.8 years; 52.7% (6,217/11,808) male) from whom 3,107 died during the 10.6-year follow-up. The CNN demonstrated excellent accuracy for retrieval of the 14 radiographic findings with area-under-the-curve (AUC) ranging between 0.83–1.00 (F1 score 0.84–0.97). Cox model showed the time-dependent AUC for predicting ACM was 0.747 (95% confidence interval [CI] of 0.704–0.790) at 30 days.

## Conclusion

An ML-based NLP approach to unstructured CT reports demonstrates excellent accuracy for the extraction of predetermined radiographic findings, and provides prognostic value in HF patients.

## Introduction

Heart failure (HF) is a complex chronic condition that is associated with significant morbidity and mortality [1]. Such increased utilization of healthcare resources through frequent medical encounters (either inpatient or outpatient) and performance of diagnostic radiologic studies typically results in the generation of a significant amount of data in the form of unstructured reports. Recently, clinical research in this field has focused on developing predictive models for the occurrence of adverse events within contemporary HF cohorts [2–4]. Unfortunately, most existent models have limited discriminatory performance as they rely on limited clinical or socio-economic variables that are structured and readily available, and as a result, fail to capture the complexity of such a disease process through incorporation of data that is available in unstructured radiology reports.

The process of data collection and extraction from electronic medical records (EMRs) for such a comprehensive approach can be arduous and time-consuming. Imaging techniques, in particular, typically do not yield structured reports; rather, radiologists dictate or enter free text reports detailing their findings into the EMR. Such a workflow necessitates advanced computational techniques to render the non-structured data computable and amenable to statistical analyses. Natural Language Processing (NLP) is a branch of computer science that focuses on developing computational models for understanding natural (human) language [5,6]. NLP has been increasingly used to automate information extraction from EMRs to streamline data collection for clinical and research purposes [7–9]. Recently, deep learning architectures, especially the ones based on Recurrent Neural Network (RNN) have been shown to exhibit good performance for clinical NLP tasks. For instance, Long Short-Term Memory (LSTM) and Convolutional Neural Network (CNN) have been used for sentiment analysis and sentence classification, and in the clinical realm an Artificial Neural Network (ANN), Context-LSTM-CNN, that combines the strength of LSTM and CNN with a context encoding algorithm was used to determine the presence of a drug-related adverse event from medical reports [10–12]. Further, LSTM was successfully used for the classification of relations within clinical notes [13].

In the present manuscript, we investigate a CNN based NLP pipeline to convert unstructured thoracoabdominal computed tomography (CT) reports into a machine-readable structured format for utilization in outcomes research. We demonstrate that with reasonably limited manual efforts, our NLP pipeline can generate structured dataset of the magnitude of order of big data to facilitate clinical research, which is essential in cohorts of patients with complex and multifactorial disease pathologies. Additionally, we show that such data extracted provides incremental prognostic information for the prediction of all-cause mortality (ACM) in a cohort of HF patients.

## Methods

### Patient population

Thoracoabdominal CT reports were obtained from the Clever-Heart registry, which is a single-center registry created to predict outcomes in HF patients. The Weill Cornell Medicine Institutional Review Board (WCM IRB) approved this study and the written informed consent for participants was waived since the registry is retrospective and did not require patient contact or disclosure of identifying information. The registry includes both structured clinical data and unstructured free-text reports. The registry consists of 21,311 patients who were admitted to the New York-Presbyterian Hospital/Weill Cornell Medicine (NYPH/WCM) and discharged with a billing diagnosis of HF (defined by an ICD-9 code of 428.* or an ICD-10 code of I50*) between January of 2008 and July of 2018. The cohort was extracted from the EMR using the Architecture for Research Computing (ARCH) groups Secondary Use of Patients' Electronic Records (SUPER) landing zone. As detailed in previous work (10), SUPER aggregates data from multiple electronic resources that are in use at NYPH/WCM. For this analysis, exclusion criteria included patients without thoracoabdominal CT reports (n = 8,788) and age <18 at the time of CT acquisition (n = 715). As a result, 11,808 patients were included in the final analysis (Fig 1). The occurrence of death was determined using data extracted from SUPER and was defined as one of the following: (1) death as determined by in-hospital mortality, (2) death as recorded in EMR, (3) an instance of an autopsy report in EMR, or (4) death as recorded in the Social Security Death Master File. Each patient's most recent follow-up was also calculated using data from SUPER and the various sources from which it draws, including both inpatient and outpatient EMRs. The most recent CT report, including the closest CT report prior to a death event, was selected for each individual.

### Choice of radiographic findings

The investigation included a combination of 14 common findings in HF patients as well as commonly reported findings on thoracoabdominal CT reports. These features were: (1) aortic aneurysm, (2) ascites, (3) atelectasis, (4) atherosclerosis, (5) cardiomegaly, (6) enlarged liver, (7) gall bladder wall thickening, (8) hernia, (9) hydronephrosis, (10) lymphadenopathy, (11) pleural effusion, (12) pneumonia, (13) previous surgery, and (14) pulmonary edema. 1,560 CT randomly selected reports were manually annotated by an experienced cardiologist (S.J.A.) for the presence or absence of these 14 radiographic findings, in addition to age and gender. Thereafter, a randomly chosen set representing 15% of the total annotations (225/1500) by physician 1 were independently validated by physician 2 with 3 years of C.T. experience (A.V. R.). The inter-observer variability was less than 2%.

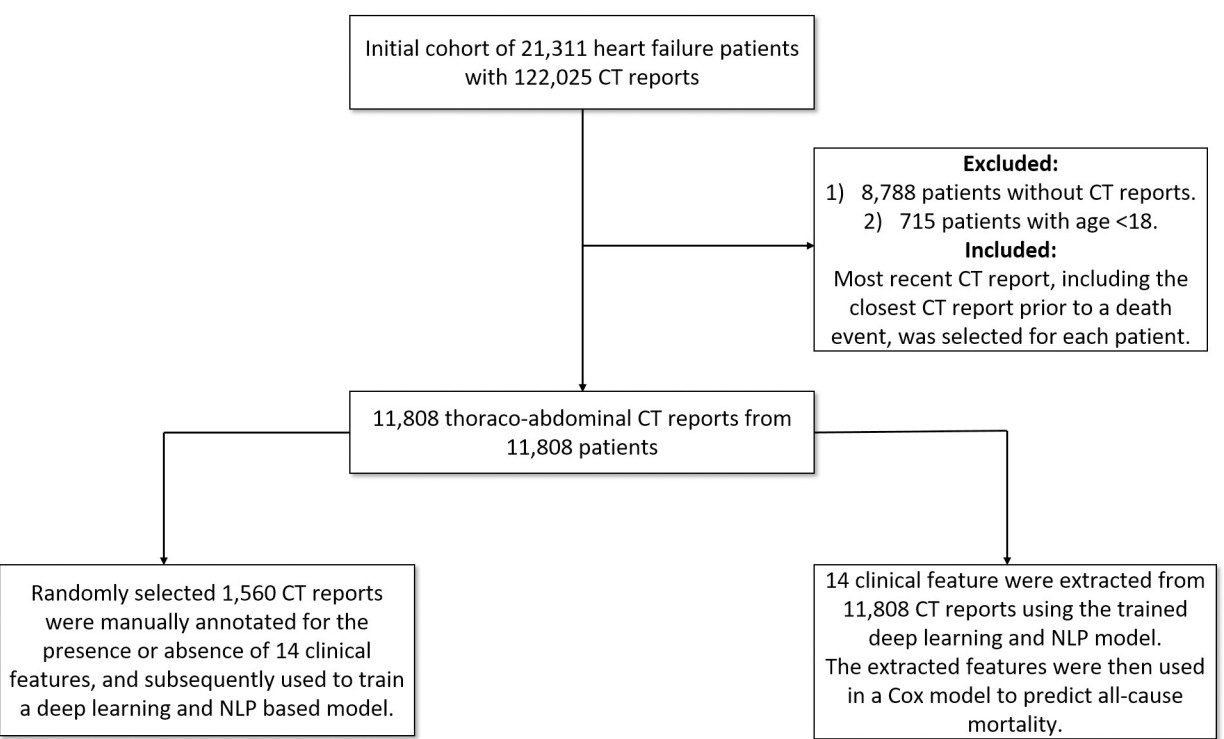

**Fig 1. Overall study design and workflow.** 11,808 thoracoabdominal CT reports from 11,808 patients were included in the final analysis.

## Rule-based NLP for determination of ground truth

Rule-based methods for tasks involving unstructured data perform well for many tasks, especially when there is syntactical uniformity in text and low level of lexical variation [11]. To develop a rule-based approach to extract our target features, a simple ruleset was developed using the manually annotated features from the 1,560 CT reports. This ruleset was based on "phrase-matching" against the corpus of reports: for example, the simple presence of the phrase *"pleural effusion"* was considered evidence for the presence of the feature "pleural effusion." This approach was based on the assumption that the free-text report would mention a particular finding only if it was present in some measure. However, there are several problems with this approach, including both the relatively high prevalence of negated mentions *(i.e. Pleural effusion is not present)* as well as lexical variations, including typographical errors *(i.e. Pleural effusion present in . . .)*. Though not very accurate, this provided a good starting place for the generation of ground truths. In addition to the 14 radiographic findings, simple regular expression rules were also identified to extract age (date of birth) and gender. Age and gender extraction were fairly accurate owing to the uniformity of age and gender syntax on CT reports.

## Development of Word2vec

Word2vec is an effective way to obtain distributed vector representations of words given a specific vocabulary [12]. It has shown to be very powerful in learning precise syntactic and semantic word relationships. Skip-gram models are used to find word representations by predicting the surrounding words, given a central word [13]. It is imperative to train word-vector representations on a corpus consisting of medical literature, to avoid a significant number of

out-of-vocabulary medical words that may appear in the data. Training Word2Vec typically requires medical corpora that are both large and diverse. Since our corpus comprised thoracoabdominal CT reports, we utilized a pre-trained word2vec as detailed by Pyysalo et al [14]. This model was trained using articles from PubMed and PMC texts. Word2Vec was trained using a skip-gram model with a window size of 5. Hierarchical softmax training was employed, with a frequent word-subsampling threshold of 0.001 to create 200 dimensional vectors [15].

## CNN model pipeline

The task of retrieving clinical terms from CT reports was treated as a multi-class, multi-label sequence classification task [16]. On average, 67% of each report contained clinical information while the remaining 33% contained patient identifiers. Accordingly, a redaction module was designed to ensure the utilization of only the relevant clinical information. The redaction module assumes that the header and footer of the reports are dedicated principally to patient identifiers. This module trims down the reports by removing the first and last $n$ lines, defining $n$ by scanning through a small vocabulary of start words of radiologic reports to determine the number of lines to be redacted. In a similar fashion, the footers were also removed. The output of the redaction module was then passed to the filtering module for the removal of special characters while retaining alphanumeric characters (S1 Fig).

The annotated 1,560 CT reports were split into training and testing sets at a ratio of 80%-20%. The training set was further split into training and validation sets, also at a ratio of 80%-20%. As proof of the generalizability of the deployed CNN architecture (S2 Fig), the test set was held to ensure the validity of the performance metrics. The validation set was used for a grid search to tune the hyper-parameters of the CNN model. We chose filter sizes of 3, 4 and 5 with 128 filters for each size. To reduce overfitting, we used dropout with 0.5 keep probability. In addition, we also used l2-regularization on the weights, with a value of lambda as 0.01. We trained each model for 50 epochs and chose the one with the highest categorical accuracy on the validation set. We optimized the categorical cross-entropy loss utilizing Adam optimizer with learning rate set to 0.001.

To benchmark the performance of the proposed approach, a classifier was trained by first converting the output of the filter module into word embedding using Term frequency-inverse document frequency (TF-IDF), then training a Naive Bayes and a Support Vector Machine (SVM) classifiers (considered as traditional machine learning algorithms). Python's scikit-learn library was used to train the benchmark model.

## Performance metrics and outcomes investigated

Binary classifiers for each clinical finding were evaluated using the Receiver Operator Characteristic (ROC) curve approach. A ROC curve is a plot of false-positive rates and true positive rates charted over a range of decision thresholds. A purely random guessing classifier would be near the $y = x$ line on the ROC plot, while a perfect classifier would look like a unit-step function for $0 < x < 1$ and will have Area Under Curve (AUC) of 1. We also report F1-score, precision and recall as additional robust metrics given the imbalanced dataset that was available.

## Prediction explanation using layer-wise relevance propagation

Bach et al. [17] introduced a layer-wise relevance propagation (LRP) in a feed-forward network to explain pixel-level classification decisions in CNNs. Starting with the output layer, LRP proceeds layer by layer and assigns the relevance of the target neuron to other units of the layers. They proposed two relevance propagation rules for the layer, assuming that the lower

layer neurons that mostly contribute to the activation of the higher layer neuron receive a higher relevance. While the convolutional network shows promising performance, the connection between the CNN learned features and the clinical findings might be difficult to see. By visualizing the contribution of specific words toward the classification task by the network proposed by Bach et al and Ancona et al, [17,18] one can identify more clearly the relationships derived by the CNN.

## Statistical analysis

Continuous variables were calculated and reported as mean ± standard deviation, whereas categorical variables were calculated and reported as counts and percentages. A Cox proportional hazards model was used for constructing survival models [19]. Backward selection based on the Akaike information criterion (AIC) was utilized for variable selection. A random survival forest model was also used for predicting ACM, defined as death from any cause [17,20]. The inverse probability of censoring weights was used to deal with right-censored survival data [21–23]. In addition, the outcome of the survival analysis was time-to-event for both the Cox model and the random survival forest model. Time-dependent AUCs were used to evaluate model performance. Further, feature importance was assessed using VIMP (variable importance) [24]. In the present scenario, VIMP in random forests for a feature, $x_a$, is the difference between prediction error when $x_a$ is noised up by permuting its value randomly, compared to prediction error under the original predictor. A p-value of less than 0.05 was considered statistically significant. All analyses were performed using R software (RStudio, Boston, MA) [25].

## Results

### Dataset

The mean word-length of CT reports was 3,085 words, with an average length of clinical information of 1,330 words (43%). Table 1 shows the prevalence of a positive finding for each of the 14 radiographic findings. While some features had a balanced prevalence of positive and negative occurrences (for example, previous surgery and atelectasis), other features exhibited a

**Table 1. Prevalence of radiographic findings between the ground truth cohort (manually annotated reports) and the Clever-Heart cohort.** For the ground truth corpus, the prevalence percentage is calculated based on known, manually annotated ground truths. For Clever-Heart, values are based on predictions from the CNN model.

| No. | Radiographic finding | Ground Truth CT reports | | Clever-Heart (As Predicted) | |
|---|---|---|---|---|---|
| | | Radiographic finding Count | Prevalence (%) | Radiographic finding Count | Prevalence (%) |
| 1 | Aortic Aneurysm | 120 | 7.69 | 754 | 6.38 |
| 2 | Ascites | 299 | 19.16 | 1516 | 12.83 |
| 3 | Atelectasis | 691 | 44.29 | 5625 | 47.63 |
| 4 | Atherosclerosis | 620 | 39.74 | 5639 | 47.75 |
| 5 | Cardiomegaly | 390 | 25.00 | 3374 | 28.57 |
| 6 | Enlarged Liver | 86 | 5.50 | 162 | 1.37 |
| 7 | GB Thickening | 25 | 1.60 | 9 | .0007 |
| 8 | Hernia | 366 | 23.46 | 2594 | 21.96 |
| 9 | Hydronephrosis | 65 | 4.16 | 136 | 1.15 |
| 10 | Lymphadenopathy | 484 | 31.02 | 2593 | 21.95 |
| 11 | Pleural Effusion | 673 | 43.14 | 4823 | 40.84 |
| 12 | Pneumonia | 278 | 17.82 | 1678 | 39.96 |
| 13 | Previous Surgery | 778 | 49.87 | 4719 | 39.96 |
| 14 | Pulmonary Edema | 94 | 6.02 | 516 | 4.36 |

significant imbalance between positive and negative findings. Gall bladder thickening, enlarged liver, pulmonary edema, and aortic aneurysm were positive in less than 10% of the reports.

## Performance metric for radiographic finding extraction

For the held-out test set of 312 CT reports, the average AUC was 0.97 across the 14 radiographic findings using the CNN architecture (average F1 score of 0.90). By comparison, Naive Bayes and SVM had an average AUC of 0.69 and 0.87 respectively (average F1 score of 0.71 and 0.67, respectively) (Table 2 and Fig 2). With regards to the individual 14 radiographic findings, the CNN model outperformed the other models on each radiographic finding. Furthermore, the redaction module for de-identification improves the AUC for 13 of the 14 radiographic findings. For the radiographic finding "aortic aneurysm", disabling the redaction module produced a slightly better AUC. Fig 3 shows examples of CT reports highlighting words that guided the CNN model towards or away from predicting the presence of a certain radiographic finding.

## NLP pipeline for prognostication in HF

The proposed CNN based NLP pipeline was subsequently used to extract these 14 radiographic findings from thoracoabdominal CT reports from 11,808 HF patients in the Clever-Heart registry (mean age of the study cohort 72.8 ± 14.8 years; 52.7% were male). The inter-quartile range of follow-up was between 129 and 1,521 days, with a median follow-up of 606 days. 3,107 death events were observed during the follow-up period. In total, 11,808 CT reports were analyzed, with 9,378 used as the training set and 2,430 used as the test set for ACM prediction. The 14 radiographic findings in addition to age and gender were included in Cox models and random survival forest. The prevalence of the 14 extracted features in this cohort is shown in Table 1. Using Cox modeling, there were significant associations between the 14 extracted radiographic findings and ACM (Fig 4 and S1 Table). For instance, the presence of pleural effusion (hazard ratio [HR] 1.63, 95% confidence interval [CI] 1.48–1.78, p<0.001) and ascites (HR 1.68, 95% CI 1.51–1.88, p<0.001) showed the strongest association with

**Table 2. Performance of the Convolutional Neural Network (CNN) model benchmarked against machine learning algorithms for the extraction of 14 pre-selected radiographic findings.**

| Feature | Naive Bayes | | | | SVM | | | | CNN | | | |
|---|---|---|---|---|---|---|---|---|---|---|---|---|
| | Precision | Recall | F1 Score | ROC AUC | Precision | Recall | F1 Score | ROC AUC | Precision | Recall | F1 Score | ROC AUC |
| Aortic Aneurysm | 0.82 | 0.91 | 0.86 | 0.49 | 0.82 | 0.91 | 0.86 | 0.94 | 0.94 | 0.94 | 0.91 | 0.98 |
| Ascites | 0.62 | 0.78 | 0.69 | 0.73 | 0.64 | 0.8 | 0.7 | 0.9 | 0.87 | 0.87 | 0.84 | 0.98 |
| Atelectasis | 0.71 | 0.57 | 0.44 | 0.8 | 0.31 | 0.56 | 0.4 | 0.85 | 0.97 | 0.96 | 0.96 | 0.98 |
| Atherosclerosis | 0.55 | 0.58 | 0.45 | 0.77 | 0.34 | 0.58 | 0.43 | 0.83 | 0.9 | 0.88 | 0.89 | 0.98 |
| Cardiomegaly | 0.56 | 0.75 | 0.64 | 0.65 | 0.52 | 0.72 | 0.6 | 0.88 | 0.86 | 0.86 | 0.84 | 0.98 |
| Enlarged Liver | 0.89 | 0.95 | 0.92 | 0.65 | 0.89 | 0.95 | 0.92 | 0.92 | 0.94 | 0.97 | 0.95 | 0.96 |
| Gall Bladder Wall Thickening | 0.97 | 0.99 | 0.98 | 0.43 | 0.98 | 0.99 | 0.99 | 0.77 | 0.97 | 0.99 | 0.98 | 0.83 |
| Hernia | 0.55 | 0.74 | 0.63 | 0.69 | 0.58 | 0.76 | 0.66 | 0.92 | 0.97 | 0.97 | 0.97 | 0.99 |
| Hydronephrosis | 0.91 | 0.95 | 0.93 | 0.68 | 0.88 | 0.94 | 0.91 | 0.88 | 0.89 | 0.95 | 0.92 | 0.99 |
| Lymphadenopathy | 0.46 | 0.68 | 0.55 | 0.75 | 0.41 | 0.64 | 0.5 | 0.82 | 0.86 | 0.85 | 0.84 | 0.95 |
| Pleural Effusion | 0.76 | 0.57 | 0.43 | 0.84 | 0.35 | 0.59 | 0.44 | 0.9 | 0.96 | 0.96 | 0.96 | 0.98 |
| Pneumonia | 0.69 | 0.83 | 0.75 | 0.67 | 0.69 | 0.83 | 0.75 | 0.88 | 0.87 | 0.88 | 0.86 | 0.96 |
| Previous Surgery | 0.73 | 0.73 | 0.73 | 0.84 | 0.23 | 0.48 | 0.31 | 0.78 | 0.86 | 0.85 | 0.85 | 0.95 |
| Pulmonary Edema | 0.88 | 0.94 | 0.91 | 0.65 | 0.88 | 0.94 | 0.9 | 0.95 | 0.93 | 0.92 | 0.89 | 1 |

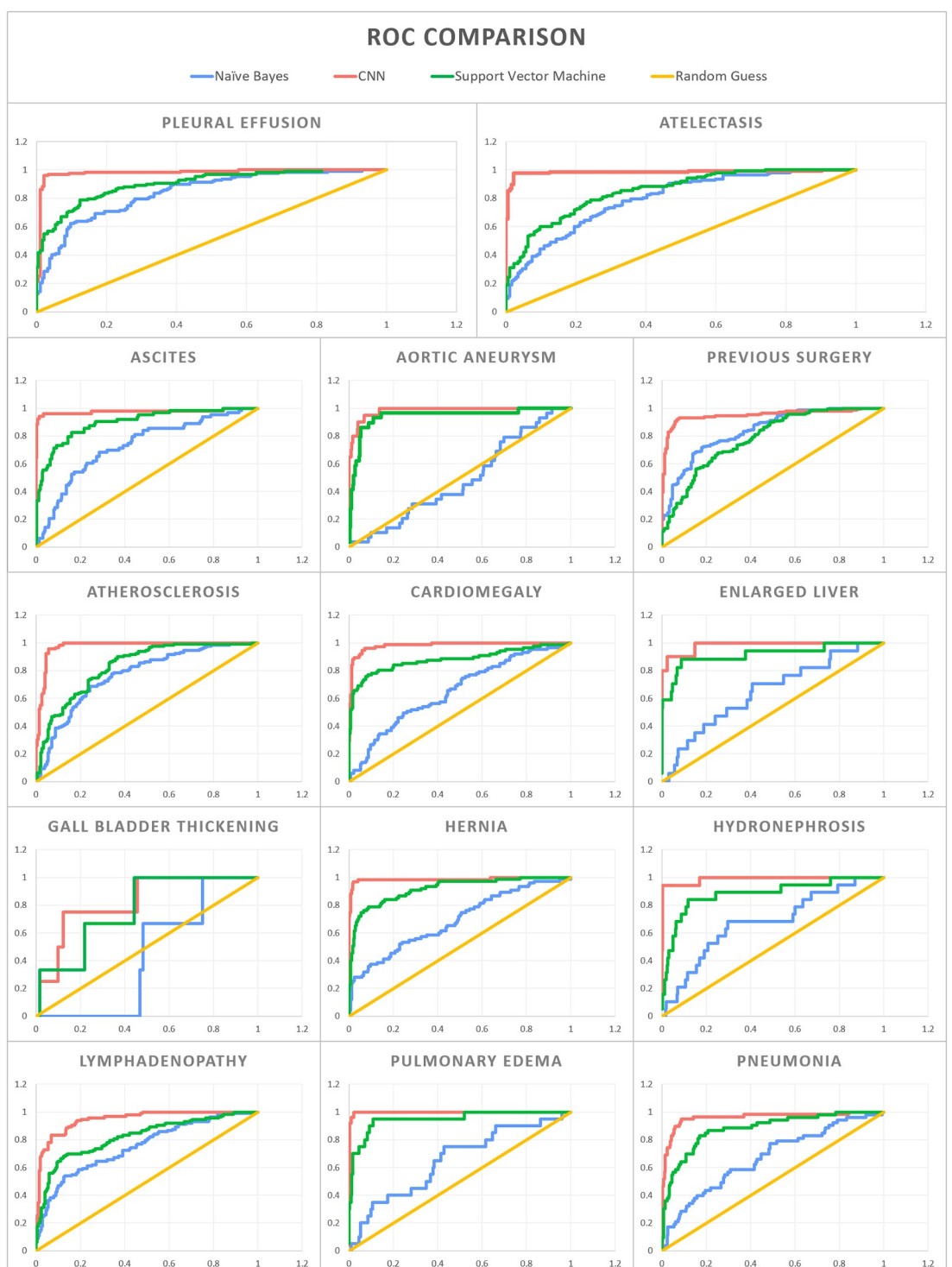

**Fig 2. Receiver Operator Characteristic (ROC) curves for the 14 pre-selected radiographic findings.** The Convolutional Neural Network (CNN) is compared to Naive Bayes, Support Vector Machine (SVM), and random guessing.

Predicted Pleural Effusion: Present

findings a dual lead pacemaker is in place the heart is normal in size there is advanced calcified atherosclerotic disease of the coronary arteries and thoracic aorta the aorta is normal in caliber there is a focal ulcerated plaque at the aortic arch without evidence of aortic dissection no large central pulmonary embolism is identified there is no pericardial effusion there is no significant thoracic lymphadenopathy there is a small right pleural effusion there is no left pleural effusion an indeterminate 5 mm pulmonary nodule is identified at the left lung apex several clusters of tree in bud pattern opacities are identified in the periphery of the right upper lobe right middle lobe left upper lobe and left lower lobe these likely reflect an infectious or inflammatory process there is compressive atelectasis in the right lower lobe the trachea and central airways are patent evaluation of the partially visualized noncontrast upper abdomen reveals cholelithiasis and is otherwise unremarkable there are no suspicious osseous lesions impression 1 advanced calcified atherosclerotic disease of the coronary arteries and thoracic aorta normal caliber aorta 2 small right pleural effusion 3 indeterminate 5 mm pulmonary nodule at the left lung apex one year ct followup is recommended to assess stability 4 cluster tree in bud pattern opacities in the periphery of both lungs likely reflect an infectious or inflammatory process 5 cholelithiasis

Predicted Atelectasis: Present

findings a dual lead pacemaker is in place the heart is normal in size there is advanced calcified atherosclerotic disease of the coronary arteries and thoracic aorta the aorta is normal in caliber there is a focal ulcerated plaque at the aortic arch without evidence of aortic dissection no large central pulmonary embolism is identified there is no pericardial effusion there is no significant thoracic lymphadenopathy there is a small right pleural effusion there is no left pleural effusion an indeterminate 5 mm pulmonary nodule is identified at the left lung apex several clusters of tree in bud pattern opacities are identified in the periphery of the right upper lobe right middle lobe left upper lobe and left lower lobe these likely reflect an infectious or inflammatory process there is compressive atelectasis in the right lower lobe the trachea and central airways are patent evaluation of the partially visualized noncontrast upper abdomen reveals cholelithiasis and is otherwise unremarkable there are no suspicious osseous lesions impression 1 advanced calcified atherosclerotic disease of the coronary arteries and thoracic aorta normal caliber aorta 2 small right pleural effusion 3 indeterminate 5 mm pulmonary nodule at the left lung apex one year ct followup is recommended to assess stability 4 cluster tree in bud pattern opacities in the periphery of both lungs likely reflect an infectious or inflammatory process 5 cholelithiasis

Predicted Pneumonia: Present

findings a left internal jugular central venous catheter is seen with tip in the right atrium a left chest wall pacemaker remains in place a nasogastric tube is seen with tip in the stomach the patient has undergone interval median sternotomy a small retrosternal fluid collection measuring 2 10 9 cm is noted and may represent small postoperative hematoma collection there is a new small to moderate pericardial effusion a new small to moderate left pleural effusion is noted a small right pleural effusion is unchanged bilateral dependent atelectasis is noted no definite focal airspace consolidation is seen to suggest pneumonia no aggressive osseous lesion is seen impression 1 status post median sternotomy with postsurgical changes including a small retrosternal fluid collection 2 new small to moderate pericardial effusion and small to moderate left pleural effusion stable small right pleural effusion the remainder of the findings in the concurrently obtained ct of the abdomen and pelvis will be dictated under a separate report

Predicted Previous Surgery: Present

findings a left internal jugular central venous catheter is seen with tip in the right atrium a left chest wall pacemaker remains in place a nasogastric tube is seen with tip in the stomach the patient has undergone interval median sternotomy a small retrosternal fluid collection measuring 2 10 9 cm is noted and may represent small postoperative hematoma collection there is a new small to moderate pericardial effusion a new small to moderate left pleural effusion is noted a small right pleural effusion is unchanged bilateral dependent atelectasis is noted no definite focal airspace consolidation is seen to suggest pneumonia no aggressive osseous lesion is seen impression 1 status post median sternotomy with postsurgical changes including a small retrosternal fluid collection 2 new small to moderate pericardial effusion and small to moderate left pleural effusion stable small right pleural effusion the remainder of the findings in the concurrently obtained ct of the abdomen and pelvis will be dictated under a separate report

Predicted Pulmonary Edema: Absent

findings a dual lead pacemaker is in place the heart is normal in size there is advanced calcified atherosclerotic disease of the coronary arteries and thoracic aorta the aorta is normal in caliber there is a focal ulcerated plaque at the aortic arch without evidence of aortic dissection no large central pulmonary embolism is identified there is no pericardial effusion there is no significant thoracic lymphadenopathy there is a small right pleural effusion there is no left pleural effusion an indeterminate 5 mm pulmonary nodule is identified at the left lung apex several clusters of tree in bud pattern opacities are identified in the periphery of the right upper lobe right middle lobe left upper lobe and left lower lobe these likely reflect an infectious or inflammatory process there is compressive atelectasis in the right lower lobe the trachea and central airways are patent evaluation of the partially visualized noncontrast upper abdomen reveals cholelithiasis and is otherwise unremarkable there are no suspicious osseous lesions impression 1 advanced calcified atherosclerotic disease of the coronary arteries and thoracic aorta normal caliber aorta 2 small right pleural effusion 3 indeterminate 5 mm pulmonary nodule at the left lung apex one year ct followup is recommended to assess stability 4 cluster tree in bud pattern opacities in the periphery of both lungs likely reflect an infectious or inflammatory process 5 cholelithiasis

Predicted Ascites: Absent

findings small right pleural effusion noted with bibasilar dependent atelectasis no pericardial effusion evaluation of the solid and hollow viscera is limited secondary to lack of intravenous and oral contrast the common bile duct spleen pancreas and adrenal glands are unremarkable segment 4a 1 8 cm hypodensity is not significantly changed from prior exam likely a hepatic cyst gallstones are seen within the gallbladder there is marked interval improvement in previously noted bilateral hydronephroureter status post placement of percutaneous nephroureteral stents a right lower quadrant ileal conduit is noted no loculated intra abdominal or pelvic fluid collection identified the small bowel is unremarkable no abdominal or pelvic lymphadenopathy or ascites is identified patient is status post cystectomy there is apparent wall thickening involving the rectum and distal sigmoid colon prominent amount of retained fecal material seen throughout the colon no suspicious osseous lesions are identified bilateral total hip arthroplasties are noted resulting streak artifact limits evaluation of the lower pelvis impression 1 interval improvement in bilateral hydronephroureter status post nephroureteral stent placement 2 status post cystectomy with right lower quadrant ileal conduit 3 apparent wall thickening of rectosigmoid colon suggesting colitis nonspecific 4 small right pleural effusion

**Fig 3. Explaining the output of a trained CNN model using layer-wise relevance propagation.** The predicted label is considered as the true class label. The color intensities are normalized to the absolute value of maximum relevance score per report such that the deepest red denotes the word with the highest positive relevance in the class label prediction, while the deepest blue denotes the most negative relevance score in the prediction of the same label.

ACM, thus indicating that the presence of such a clinical finding led to higher mortality. Similarly, random survival forest also demonstrated that the presence of pleural effusion and ascites were most highly correlated with mortality.

Cox survival models were constructed using features extracted from the developed CNN-based pipeline and compared to that of non-deep learning-based featurization using Naïve Bayes modeling (Fig 5). The CNN-based NLP pipeline for unstructured text featurization yielded enhanced clinical risk prediction, compared to Naïve Bayes modeling for outcomes at 30 days (AUC of 0.747 vs. 0.604, respectively; p<0.01), 60 days (AUC of 0.758 vs. 0.625, respectively; p<0.01) and 365 days (AUC of 0.739 vs. 0.598, respectively; p<0.01). Further, the use of radiographic findings extracted using the CNN-based NLP pipeline resulted in the enhanced prediction of outcomes across multiple survival models. The full Cox model showed the time-dependent AUC for predicting ACM is 0.747 (95% CI of 0.704–0.790) at 30 days (C-statistic of 0.695 ± 0.012), 0.758 (95% CI of 0.720–0.796) at 60 days and 0.739 (95% CI of 0.708–0.770) at 1 year. The Cox model with select features (pneumonia, pleural effusion, lymphadenopathy,

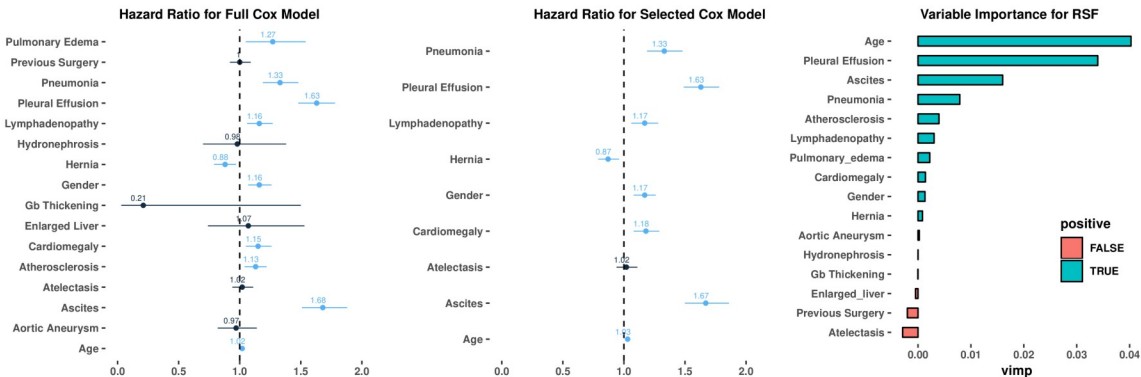

**Fig 4. Correlation between radiographic findings and all-cause mortality.** (A-B): Forest plot for the Cox model with all 14 variables and selected 9 variables. Numbers represent hazard ratios. The range of lines indicates 95% confidence intervals. Color blue implies significant variables with p<0.05. (C) Variable importance plot of the Random Survival Forest model. Large positive values indicate the dependency of the outcome to get high predictive power. Values closer to zero represent a lower contribution to improved predictive accuracy. Negative numbers indicate the predictive accuracy would improve when the variables were unspecified.

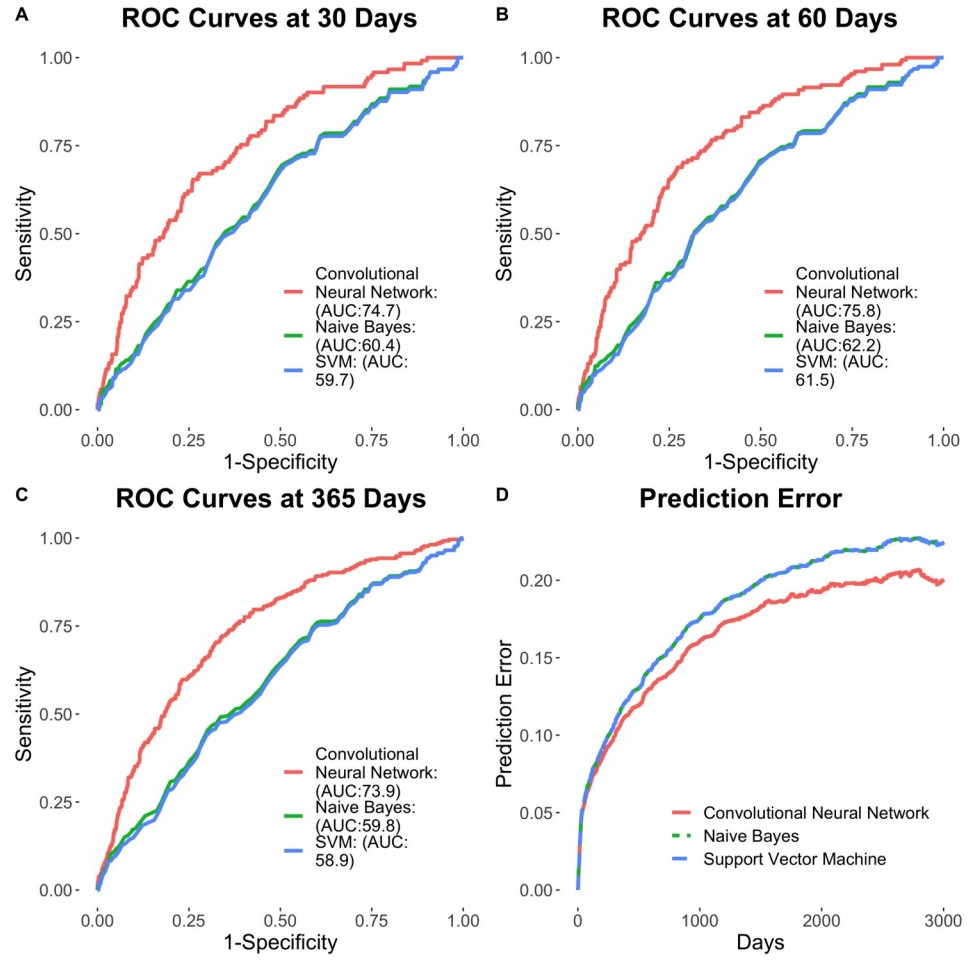

**Fig 5. Prognostication of outcomes using the CNN model.** (A-C) Time-dependent ROC curves at 30, 60 and 365 days. (D) Time-dependent Brier scores.

hernia, cardiomegaly, atelectasis, ascites, gender, and age) according to AIC showed similar time-dependent AUC results: 0.745 (95% CI of 0.701–0.788) at 30 days, 0.756 (95% CI of 0.719–0.795) at 60 days and 0.738 (95% CI of 0.706–0.769) at 1 year. On the other hand, the random survival forest model performed worse than the Cox models, with time-dependent AUC of 0.701 (95% CI of 0.659–0.744) at 30 days, 0.687 (95% CI of 0.649–0.725) at 60 days and 0.670 (95% CI of 0.638–0.702) at 1 year. Similar results can be observed in the Brier score plot, in which random survival forest exhibited the highest prediction error and the two Cox models performed in a similar fashion, with lower error (S3 Fig).

## Discussion

The present investigation highlights the potential of an ML framework that applies NLP to unstructured thoracoabdominal CT reports in the accurate extraction of pre-specified radiographic findings as well as a prognostication of outcomes using the extracted clinical data. While the lack of a standard syntactical structure for free text reports across different sites makes it difficult to retrieve clinical information using traditional rule-based approaches and regular expressions, our analysis shows that the use of a CNN resulted in the accurate extraction of specific clinical terms. The AUC for radiographic finding extraction ranged from 0.88 to 1.0, with an average AUC of 0.96. The study also showed that the CNN model had significantly improved performance when trained against carefully preprocessed input text: removal of headers and footers from the CT reports boosted performance metrics for 13 out of the 14 radiographic findings. In addition, the extracted clinical information was found to be useful for prognostication of outcomes, specifically with the occurrence of ACM, within an HF cohort. Furthermore, such an approach could be curtailed for future clinical applications such as stratification of HF individuals at risk for hospitalizations and readmissions, which could reduce the significant healthcare expenditure associated with HF management.

Free-form language in EMR is unrestricted and is subject to endless interpreter-to-interpreter and site-to-site variations. The theoretical potential for lexical variation in the expression of a given concept is infinite—hence, interpretation and computational modeling of context is crucial for the development of tools that go beyond simple string matching. Our work here demonstrates that a CNN based NLP approach can provide enhanced performance on these tasks by incorporating novel computational models. Illustrative examples are as in Fig 4. The CNN model predicts the presence of atelectasis by assigning the highest relevance to *"compressive atelectasis"*. Similarly, it learns that the clinical term "pneumonia" is present by assessing the highest relevance to *"pneumonia"* in the report. However, while classifying pulmonary edema, the model learns *"pulmonary"* with the highest positive relevance but at the same time assigns *"embolism"* with the highest negative relevance score, demonstrating that the model "understands" that pulmonary edema is different from a pulmonary embolism. In the case of the term "ascites", even though the model learns that the presence of this term is most important for prediction of ascites, the filters of the CNN learn negation from neighboring words. As a result, even though the report says, *"ascites is identified . . ."* the model aggregates this with the earlier negation in the report and hence was able to classify this report as ascites-absent "*no ascites is identified*". The model's ability to comprehend free-form medical text is further demonstrated in the analysis of "previous surgery" and "pleural effusion" classification. For previous surgery, *"median sternotomy"* was scored with the most positive relevance. The CNN model coupled with word2vec trained on medical corpora understands that median sternotomy is a surgical procedure even without explicit training. Words like *"post"* and *"undergone"* help the model to learn the context and deduce that the procedure was performed in the past and hence flags the report as positive for the target concept "prior surgery."

Similarly, the "pleural effusion" prediction model learns with the highest positive relevance that *"a small right pleural effusion"* was present, while simultaneously taking into consideration the presence of the term *"no left pleural effusion"*, assigning high relevance to that phrase as well. The model also assigns the highest negative relevance score to *"no pericardial effusion"* to prevent it from falsely flagging the feature of pericardial effusion as an instance of a pleural effusion. Overall, the model was able to average these different inferences about pleural effusion in this report and ultimately correctly deduce the presence of a pleural effusion.

While the inference on "atelectasis" and "pneumonia" could be achieved using string matching and simple rule-sets, our model understands more complex sentences and structures as prevalent in imaging reports. An additional benefit of this technique is the extent to which it minimizes the need for labor-intensive human effort. In recent investigations, recurrent neural networks (RNNs) have shown excellent results for text comprehension tasks—however, their need for copious amounts of annotated examples makes them unsuitable for NLP applications in the clinical field, where human annotation often requires clinical expertise, which can become a burden within a busy clinical setting. Our CNN-based method proved to effectively balance the need for both high performance and minimized requirement for manual effort (as evident by the high extraction accuracy using 1,560 annotated CT reports).

NLP has been increasingly used for clinical applications over the past few years. The Linguistic String Project—Medical Language Processor (LSP-MLP), conducted at New York University in 1987, was among the first large-scale projects using NLP within the context of clinical research [26]. The LSP-MLP sought to help physicians extract and summarize sign/symptom information, drug dosage, and response data, and to identify possible medication side effects. Those results revealed the utility of various computational linguistic experiments in the medical field. In addition, substantial efforts have been directed towards NLP pipelines to supplement or classify conditions based on ground truths provided by the International Classification of Diseases (ICD) codes. For example, Pakhomov et al. found that NLP algorithms consistently retrieve more clinical information from free-text reports, and specifically regarding the presence of angina pectoris, than what is provided with ICD codes [27]. A naïve yet powerful NLP approach in evaluating clinical texts is by studying sentence modifiers. State of the art tools, such as NegEx or NegBio look at finding possible negation to clinical entities in discharge summaries and radiology reports, respectively [28–29]. However, the application of NLP to clinical data and radiologic reports is an active area of research, since the field is in the process of establishing a framework for methodological evaluation. Contemporary practices utilize human manual annotations as the ground truth and generate performance metrics based on comparisons with the prediction models. From a clinical perspective, NLP frameworks could prove to be a valuable resource in the care of patients with complex chronic conditions such as HF. As a disease process, HF has intricate pathophysiology as well as disease manifestations, leading to a heterogeneous expression of symptoms and subsequent outcomes. HF patients undergo numerous imaging modalities as part of the initial diagnostic workup, monitoring of therapy, assessment of underlying structural progression, as well as evaluation of concomitant non-cardiovascular pathologies. Extraction of imaging findings from radiologic reports could thus be complementary to patient-level clinical information and could further help prognosticate and better risk-stratify individuals, especially in the setting of HF and similar multifaceted diseases.

While our investigation has clear advantages in terms of clinical applications and novelty of this approach, there are several limitations worthy of mentioning. Firstly, this was a single-center study of thoracoabdominal CT reports obtained from HF patients. While the CNN model had impressive accuracy for the extraction of 14 pre-selected radiographic findings, the performance of this model on free-text CT reports from other sites is unknown. Secondly, several

radiographic findings, such as gall bladder wall thickening, had a high rate of false-negative results, which is attributable to severe class imbalance since very few reports had a positive label for this class (i.e. most reports did not mention the presence of gall bladder wall thickening). A better sampling approach and the inclusion of gastrointestinal disease cases could be used in the subsequent analysis to further improve the diagnostic performance of our model. Finally, the study used the latest CT reports in the survival analysis, instead of including all available CT reports. This was done since complex modeling is required in order to utilize time-dependent covariates. The scope of the present investigation was to demonstrate the utility of CNN for clinical finding extraction from unstructured CT reports, while subsequent investigations will aim at developing sophisticated models that include time-dependent variables for prognostication of outcomes.

In summary, we show that a CNN based NLP pipeline applied to unstructured CT reports accurately extracts 14 pre-specified radiographic findings in a cohort of HF patients undergoing thoracoabdominal CT imaging, which in turn provides prognostic value for prediction of ACM in such a cohort. The approach detailed herein offers the potential to supplement the extraction of clinical data, beyond that of existing structured data in EMR systems, for outcomes research and clinical care especially for individuals with chronic and complex medical conditions such as HF.

## Supporting information

**S1 Fig. Stacked Convolutional Neural Network (CNN) architecture.** The input in the training phase consisted of 1,560 free-text thoracoabdominal computed tomography reports. These reports passed through redaction module to remove headers consisting of non-clinically relevant free text. The filtering module tokenized and removed special characters. After converting these reports into 200 dimensional vectors for each word, they were then used to train the CNN model. The output of the training phase was a trained CNN model. A separate CNN model for each of the 14 clinical findings constituted the model zoo. In the Go Live phase, input consisting of 11,808 free-text reports was passed to the trained CNN models after passing through the redaction module, the filtering module, and conversion to word vectors. The output from this phase generated the database of clinical findings which, along with age and gender, were used for prediction of all-cause mortality.
(TIF)

**S2 Fig. Structure of the Convolutional Neural Network (CNN).** (a) Filter bank with kernels of size 2, 3 and 4. Each kernel is a 200-dimensional vector which is the same as the dimensionality of the word embedding used. (b) Each word of the sentence is converted to a 200-dimensional vector using word2vec. For uniformity in length, shorter sentences are zero padded. Filters of size 2, 3 and 4 are individually convolved with the sentence matrix. (c) Feature maps generated for each filter size. (d) Max pooling over time to create single feature vector. (e) Fully connected layer with SoftMax to classify presence or absence of the specific clinical finding.
(TIF)

**S3 Fig. Prognostication of outcomes using the deep learning-based method for feature extraction.** (A-C) Time-dependent ROC curves at 30, 60 and 365 days for 3 models created using the deep learning-based method for feature extraction: (1) a full COX model, (2) a COX model with select variables and (3) a random survival forest. Panel D shows the time-dependent Brier scores.
(TIFF)

**S1 Table.** (A) Full Cox model for correlation between the radiographic finding and all-cause mortality. (B) Select Cox model and correlation between radiographic findings and all-cause mortality.
(DOCX)

## Author Contributions

**Conceptualization:** Mohit Pandey, James K. Min, Subhi J. Al'Aref.

**Data curation:** Mohit Pandey, Zhuoran Xu, Evan Sholle.

**Formal analysis:** Mohit Pandey, Zhuoran Xu, Gabriel Maliakal, Gurpreet Singh.

**Methodology:** Mohit Pandey, James K. Min, Subhi J. Al'Aref.

**Project administration:** Mohit Pandey.

**Resources:** Evan Sholle.

**Supervision:** James K. Min, Subhi J. Al'Aref.

**Validation:** Zhuoran Xu.

**Visualization:** Zhuoran Xu.

**Writing – original draft:** Mohit Pandey.

**Writing – review & editing:** Evan Sholle, Zahra Fatima, Daria Larine, Benjamin C. Lee, Jing Wang, Alexander R. van Rosendael, Lohendran Baskaran, Leslee J. Shaw.

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
