## [Decision Letter · Decision Letter 0]

22 Apr 2020

PONE-D-20-06619

Extraction of clinical features from unstructured thoracoabdominal computed tomography reports using convolutional neural network based natural language processing

PLOS ONE

Dear Dr. Al’Aref,

Thank you for submitting your manuscript to PLOS ONE. After careful consideration, we feel that it has merit but does not fully meet PLOS ONE’s publication criteria as it currently stands. Therefore, we invite you to submit a revised version of the manuscript that addresses the points raised during the review process.

We would appreciate receiving your revised manuscript by Jun 06 2020 11:59PM. To enhance the reproducibility of your results, we recommend that if applicable you deposit your laboratory protocols in protocols.io, where a protocol can be assigned its own identifier (DOI) such that it can be cited independently in the future. For instructions see: http://journals.plos.org/plosone/s/submission-guidelines#loc-laboratory-protocols

We look forward to receiving your revised manuscript.

Kind regards,

Yifan Peng, Ph.D.

Academic Editor

PLOS ONE

Additional Editor Comments:

1. Both reviewers recommend reporting metrics of precision, recall, sensitivity, specificity, and F1 for each category. In addition, the C-statistic is the most relevant metric for prognosis. It is also recommended to report C-statistic for ACM prognosis.

2. The discussion of rule-based systems needs to include SOTA tools, such as NegEx or NegBio.

3. It is not clear if the features extracted from CNN and used in Cox are clinical features or the last layer of CNN. The use of the word "feature" is confused as it can either mean the deep feature that is the output of a layer within a CNN model, or a specific feature pre-defined by the authors. To this end, I would recommend changing the "clinical features" to "radiographic findings".

2. Thank you for stating the following in the title page of your manuscript:

'Funding: This study received support from the New York-Presbyterian Hospital (NYP) and Weill Cornell Medicine (WCM), including the Clinical and Translational Science Center (CTSC) (UL1 TR000457) and Joint Clinical Trials Office (JCTO).

'The author(s) received no specific funding for this work.'

Please clarify the sources of funding (financial or material support) for your study. List the grants or organizations that supported your study, including funding received from your institution.

State what role the funders took in the study. If the funders had no role in your study, please state: “The funders had no role in study design, data collection and analysis, decision to publish, or preparation of the manuscript.”

If any authors received a salary from any of your funders, please state which authors and which funders.

If you did not receive any funding for this study, please state: “The authors received no specific funding for this work.”

'The authors have declared that no competing interests exist.'

Thank you also for stating the following in the title page of your manuscript:

'Disclosure: Dr. James K. Min receives funding from the Dalio Foundation, National Institutes of Health, and GE Healthcare. Dr. Min serves on the scientific advisory board of Arineta and GE Healthcare, and has an equity interest in Cleerly. All other authors have reported that they have no relationships relevant to the contents of this paper to disclose.'

We note that one or more of the authors are employed by a commercial company: GlaxoSmithKline

4.  Thank you for stating in the manuscript:

'Written informed consent for participants was waived by the institutional review board (IRB) since the registry is retrospective and did not require patient contact or disclosure of identifying information'

a. Please amend your current ethics statement to include the full name of the ethics committee/institutional review board(s) that approved your specific study.  Please state specifically whether the IRB approved the study or whether approval was waived.

Reviewers' comments:

Reviewer's Responses to Questions

**Comments to the Author**

1. Is the manuscript technically sound, and do the data support the conclusions?

Reviewer #1: Partly

Reviewer #2: Partly

2. Has the statistical analysis been performed appropriately and rigorously? 

Reviewer #1: Yes

Reviewer #2: Yes

3. Have the authors made all data underlying the findings in their manuscript fully available?

Reviewer #1: No

Reviewer #2: Yes

4. Is the manuscript presented in an intelligible fashion and written in standard English?

Reviewer #1: Yes

Reviewer #2: Yes

5. Review Comments to the Author

Reviewer #1: The manuscript titled “Extraction of clinical features from unstructured thoracoabdominal computed tomography reports using convolutional neural network based natural language processing” addresses an important clinical problem of predicting mortality of heart failure patients. The paper describes a substantial amount of work and the experimental designs are sound. The manuscript, though missing some background information of relevant work, is generally easy to follow. The experimental results should be further extended to include metrics that can reflect the system performances in imbalanced datasets. Overall, it is a promising study with sound design, and some parts of the manuscripts requires further revision.

I have some comments and hopefully the manuscript can be improved by addressing them:

The introduction did not include sufficient relevant study to highlight the contribution to this manuscript. How is this study different from other work in relevant questions? The authors briefly mentioned it, but it requires further details for clarity.

Rule-based mention extraction is commonly used along with contextual information (negation, certainty, subject) such as NegEx or its extensions. There are mature NLP systems (e.g. cTAKES, MedTagger) extracting features along with the contexts. Therefore the claim that rule-based systems cannot handle them (L160) is not true.

L164: why do age and gender need to be extracted from clinical notes? Are there any difficulties using structured demographic information which should be accurate?

L183-189: the pre-processing step is not relevant to the main contributions. Since there is no concrete example of the report showing why this step (redaction) is important, I suggest the authors simplify the descriptions.

Figure 2 and 4 are barely legible. Figure 2 is actually a system workflow rather than the CNN architecture, which should be Supplement Figure 1.

Figure 4 is very interesting, but how is the relevance score calculated or generated? Are there any sections or document structures in given examples?

The dataset is very imbalanced. There are several features (1,6, 7, 9, 14) that have prevalence less than 10%. It is important to report other measures such as precision, recall and F1-score to properly report the model performances. Also, for features like 7, with such low prevalence, it is hard to train a useful CNN and claim the experimental result is meaningful.

The head row of Table 2 is misaligned.

Can machine learning methods also predict mortality using NLP features at the given test time? Do the NLP features outperform other features (e.g. ICD codes) so that the feature extraction work is necessary or meaningful?

Reviewer #2: ROC is known to be biased for imbalanced datasets, which you have. For example, always predicting no aortic aneurysm would have a 92% accuracy and high AUC. This renders the results difficult to assess. This could be mitigated by either (a) reporting performance of a majority-class classifier or (b) reporting more robust metrics such as area under the precision-recall curve, precision, sensistivity, specificity, and F1-measure.

It is not clear to me how the Cox model and CNN are related.

This paper reports on the use of a convolutional neural network to extract clinical features of heart failure from unstructured clinical reports. Reports were manually annotated for mentions of 14 clinical features. The text of each report is represented as a sequence of words, wherein each word was encoded using a pre-trained Skip Gram model.

The manuscript is well organized and generally clear. However, I have some concerns with the experimental setup.

First, the study evaluates approaches based on ROC AUC. ROC AUC is known to be biased when working with imbalanced datasets. This is unfortunately because most of the clinical features have very skewed distributions. For example, the low prevalance of aortic aneurysm means that a classifier that always predicts the absense of that feature would have a 92% accuracy and proportionally high ROC AUC. This renders the results difficult to asses, greatly limiting the generalizability of the work. This issue could be mitigated by (a) reporting the performacne of a majority-class baseline classifier, and/or (b) reporting additional more robust metrics such as the area under the precision-recall curve, precision, sensistivity, specificity, and F1-measure. The following paper provides an excellent discussion on the shortcomings of ROC plots with imbalanced datasets:

Saito, T., & Rehmsmeier, M. (2015). The precision-recall plot is more informative than the ROC plot when evaluating binary classifiers on imbalanced datasets. PloS one, 10(3), e0118432. https://doi.org/10.1371/journal.pone.0118432

Second, the feature analysis is not clearly described. In general, the weights of neurons with non-linear activations are not guaranteed to be proportional to the importance of a feature. For this reason, the manuscript should provide more information on how feature weights for analysis and how the analysis was performed.

Third, the correlation to all-cause mortality is interesting, but is under-described. The authors should consider promoting it from the discussion to part of the methodology and providing more information about how the correlation is done.

There are some minor issues as well:

1. It is unusual to describe rule-based methods as "unsupervised." Rule-based approaches do not rely on machine learning: there is no training, fitting, or learning phase to supervise. Instead the decision function is directly provided through pre-defined rules.

2. On line 206, "As proof of the generalizability of the deployed CNN architecture (S1 figure), the test

set was held to ensure the validity of the performance metrics." is redundant as the test set is, by definition, witheld from training

3. On line 212, "We optimized the categorical cross-entropy loss utilizing Adam optimizer was used with a learning rate set to 0.001." is ungrammatical.

4. On line 241, ACM should be defined.

6. PLOS authors have the option to publish the peer review history of their article (what does this mean?). If published, this will include your full peer review and any attached files.

Reviewer #1: No

Reviewer #2: No

---

## [Author Response · Author response to Decision Letter 0]

11 Jun 2020

Reviewer Comments (Reviewer #1)

The manuscript titled “Extraction of clinical features from unstructured thoracoabdominal computed tomography reports using convolutional neural network based natural language processing” addresses an important clinical problem of predicting mortality of heart failure patients. The paper describes a substantial amount of work and the experimental designs are sound. The manuscript, though missing some background information of relevant work, is generally easy to follow. The experimental results should be further extended to include metrics that can reflect the system performances in imbalanced datasets. Overall, it is a promising study with sound design, and some parts of the manuscripts requires further revision.

I have some comments and hopefully the manuscript can be improved by addressing them:

The introduction did not include sufficient relevant study to highlight the contribution to this manuscript. How is this study different from other work in relevant questions? The authors briefly mentioned it, but it requires further details for clarity.

We thank the reviewer for the comment. We have added the following section to the introduction in order to highlight the strengths and contribution of our analysis: We demonstrate that with reasonably limited manual efforts, our NLP pipeline can generate structured dataset of the magnitude of order of big-data to facilitate clinical research, which is essential in cohorts of patients with complex and multifactorial disease pathologies. 

Rule-based mention extraction is commonly used along with contextual information (negation, certainty, subject) such as NegEx or its extensions. There are mature NLP systems (e.g. cTAKES, MedTagger) extracting features along with the contexts. Therefore, the claim that rule-based systems cannot handle them (L160) is not true.

We thank the reviewer for the comment. We agree with the reviewer that these mature NLP systems are able to annotate clinical datasets given the rules. However, to design all possible rules for free form medical language is not just ardent but also impractical. We remove the claim that these rule-based systems cannot handle clinical feature extraction in L160.

L164: why do age and gender need to be extracted from clinical notes? Are there any difficulties using structured demographic information which should be accurate?

We thank the reviewer for the comment. In a traditional clinical workflow, radiology reports include a short clinical summary that is provided by the clinician to the interpreting radiologist (For example: 52 male patient with abdominal pain). A radiologist has access to the electronic health record (EHR), and in challenging cases they usually revert to EHR for more clinical and imaging data, but that is the exception and not the norm (especially in a busy clinical setting). As such, in order to create a model that reflects a real-world scenario, we decided to only incorporate age and gender which are the two demographic variables that are consistently available in a radiology report.

L183-189: the pre-processing step is not relevant to the main contributions. Since there is no concrete example of the report showing why this step (redaction) is important, I suggest the authors simplify the descriptions.

We thank the reviewer for their comment. We had elected to highlight the pre-processing step for the following reasons:

1. This exercise removes a vast majority of Personally Identifiable Information (PII) in compliance to HIPPA.

2. The effects of “Garbage in, garbage out” in machine learning is well studied. There is no intuitive reason that a particular PII should be correlated to any radiological finding. If such an occurrence happens (for instance all Dr. Bob Smith’s patients in the training dataset coincidentally happened to have Hernia), the Convolutional Neural Network model might learn associating presence of hernia to presence of Dr. Bob Smith’s PII. The redaction steps prevent such a learning to take place. 

Sessions, Valerie, and Marco Valtorta. "The Effects of Data Quality on Machine Learning Algorithms." ICIQ 6 (2006): 485-498.

Since the pre-processing step removes PII, we elected not to provide an example to be in compliance with HIPPA and the journal requirements. 

Figure 2 and 4 are barely legible. Figure 2 is actually a system workflow rather than the CNN architecture, which should be Supplement Figure 1.

We thank the reviewer for the comment. We have modified the way figures 2 and 4 are uploaded in order to be higher resolution and legible to the reviewers and editor. As suggested by the reviewer, figure 2 was changed to supplement figure 1 and the rest of the images were relabeled accordingly. 

Figure 4 is very interesting, but how is the relevance score calculated or generated? Are there any sections or document structures in given examples?

We thank the reviewer for the comment. The relevance scores in figure 4 are calculated using a technique called Layer-wise Relevancy Propagation. The work is described in L288-298. Accordingly, we modified the section header to read: Prediction Explanation using Layer-wise Relevance Propagation. We normalize relevancy score as calculated by LRP for each report and plot deepest red to denote the word with the highest positive relevance in the class label prediction, while the deepest blue denotes the most negative relevance score in the prediction of the same label.

The dataset is very imbalanced. There are several features (1,6, 7, 9, 14) that have prevalence less than 10%. It is important to report other measures such as precision, recall and F1-score to properly report the model performances. Also, for features like 7, with such low prevalence, it is hard to train a useful CNN and claim the experimental result is meaningful.

We would like to thank the reviewer for their comments. We have added precision, recall and F1-score to results in the manuscript (as recommended by the editor and reviewers). We thank the reviewer for the comment regarding the low prevalence of gall bladder wall thickening and the likely limited generalizability of our CNN model. The present analysis was focused on extracting relevant radiographic findings from CT reports in heart failure patients, irrespective of prevalence of the associated findings. Further, a model that is able to detect the absence, as opposed to the presence, of such a finding would be very useful, since the absence of gall bladder wall thickening in heart failure is an important finding (the absence of extensive edema, which leads to gall bladder wall edema and thickening is highly relevant for prognosis). Nevertheless, the real-world utility of our model can only be confirmed when we externally validate our model on a separate cohort, which the investigators hope to perform in subsequent analysis. 

The head row of Table 2 is misaligned.

We thank the reviewer for the correction. Misalignment was corrected.

Can machine learning methods also predict mortality using NLP features at the given test time? Do the NLP features outperform other features (e.g. ICD codes) so that the feature extraction work is necessary or meaningful?

We thank the reviewer for the excellent comment. We expect that ICD codes would provide predictive value, as patients with malignancy will likely have worse outcome that patients with appendicitis. However, ICD codes have little pathophysiological correlation to the underlying disease state. For example, a CT of the chest might be ordered for “pulmonary embolism”, while the scan might eventually show an underlying infection “pneumonia”. Also different institutions might use different ICD codes for a certain presentation, and a model constructed on ICD codes might not generalize well to other cohorts. On the other hand, extracted radiologic findings describe a set of findings that directly describe the underlying disease process (the constellation of cardiomegaly, pulmonary edema and pleural effusions will most likely represent the occurrence of heart failure), while these findings are ubiquitous and standard across all healthcare institutions. As such, a model built upon radiologic findings, rather than ICD codes, will likely generalize well and at the same time be based on an accurate description of the underlying disease state. 

 

Reviewer Comments (Reviewer #2)

ROC is known to be biased for imbalanced datasets, which you have. For example, always predicting no aortic aneurysm would have a 92% accuracy and high AUC. This renders the results difficult to assess. This could be mitigated by either (a) reporting performance of a majority-class classifier or (b) reporting more robust metrics such as area under the precision-recall curve, precision, sensitivity, specificity, and F1-measure.

We would like to thank the reviewer for the suggestion and comments. We have added precision, recall and F1-score to results in the manuscript as suggested by the reviewer and editor.

It is not clear to me how the Cox model and CNN are related.

We thank the reviewer for the comment. We utilized the trained CNN (on 1560 annotated radiology reports) to extract radiological findings (aortic aneurysm, ascites, etc.) from unstructured, unannotated radiology reports. These unannotated reports came from 11,808 patients. The forward pass using the CNN gave us predicted radiological findings for these patients. These findings then, along with age and gender formed the basis for outcomes prediction. We used a COX model for outcomes prediction (all-cause mortality in a cohort of heart failure patients who had the unannotated CT reports). 

This paper reports on the use of a convolutional neural network to extract clinical features of heart failure from unstructured clinical reports. Reports were manually annotated for mentions of 14 clinical features. The text of each report is represented as a sequence of words, wherein each word was encoded using a pre-trained Skip Gram model. The manuscript is well organized and generally clear. However, I have some concerns with the experimental setup.

First, the study evaluates approaches based on ROC AUC. ROC AUC is known to be biased when working with imbalanced datasets. This is unfortunately because most of the clinical features have very skewed distributions. For example, the low prevalence of aortic aneurysm means that a classifier that always predicts the absence of that feature would have a 92% accuracy and proportionally high ROC AUC. This renders the results difficult to assess, greatly limiting the generalizability of the work. This issue could be mitigated by (a) reporting the performance of a majority-class baseline classifier, and/or (b) reporting additional more robust metrics such as the area under the precision-recall curve, precision, sensitivity, specificity, and F1-measure. The following paper provides an excellent discussion on the shortcomings of ROC plots with imbalanced datasets:

Saito, T., & Rehmsmeier, M. (2015). The precision-recall plot is more informative than the ROC plot when evaluating binary classifiers on imbalanced datasets. PloS one, 10(3), e0118432. https://doi.org/10.1371/journal.pone.0118432

We would like to thank the reviewer for the comment. We have added precision, recall and F1-score to results in the manuscript.

Second, the feature analysis is not clearly described. In general, the weights of neurons with non-linear activations are not guaranteed to be proportional to the importance of a feature. For this reason, the manuscript should provide more information on how feature weights for analysis and how the analysis was performed.

We would like to thank the reviewer for their comment. The 14 features that were initially manually annotated, used to train a CNN model, and thereafter extracted by the CNN model from unseen radiology reports for prognostication were determined to be relevant findings in radiology reports by expert clinicians (these findings are typically present in heart failure patients). As such, the investigators determined that it would be clinically useful to have such a method to automatically extract such features from radiology reports and create a data repository. The primary objective following the creation of a large dataset for the clinically relevant radiological features (as described), was to demonstrate that these features and hence our proposed NLP pipeline can facilitate outcomes research. A more in-depth feature analysis of CNN is beyond the scope of our current work as we intended to use all these features regardless as they are clinically relevant to the outcome. We have added a section on how correlation between features and the outcome was performed, as explained in the next comment (see below).

Third, the correlation to all-cause mortality is interesting, but is under-described. The authors should consider promoting it from the discussion to part of the methodology and providing more information about how the correlation is done.

We would like to thank the reviewer for their comment. We have added the following paragraph to the methods section: Further, variable importance was assessed using VIMP (variable importance) [24]. In the present scenario, VIMP in random forests for a feature, xa, is the difference between prediction error when xa is noised up by permuting its value randomly, compared to prediction error under the original predictor.

There are some minor issues as well:

1. It is unusual to describe rule-based methods as "unsupervised." Rule-based approaches do not rely on machine learning: there is no training, fitting, or learning phase to supervise. Instead the decision function is directly provided through pre-defined rules.

We would like to thank the reviewer for their comment. We have made modification to the manuscript. We removed the line “Moreover, they are generally unsupervised techniques, only requiring human annotation for formal validation of their output”.

2. On line 206, "As proof of the generalizability of the deployed CNN architecture (S1 figure), the test

set was held to ensure the validity of the performance metrics." is redundant as the test set is, by definition, withheld from training.

We would like to thank the reviewer for their comment. In the said line, we would like to emphasize that to indicate generalizability of the model on unseen data we utilized a held-out test set and did not perform a k-fold cross validation approach for the same. We agree with the reviewer that it is redundant, but for a clinical audience that is not familiar with such an approach we believe that it is important to explicitly state in order to avoid any confusion.

3. On line 212, "We optimized the categorical cross-entropy loss utilizing Adam optimizer was used with a learning rate set to 0.001." is ungrammatical.

We would like to thank the reviewer for their comment. We have corrected the grammatical structure of the sentence.

4. On line 241, ACM should be defined.

We thank the reviewer for the comment. ACM (all-cause mortality) was defined as death from any cause.  

Again, we greatly appreciate the reviewer’s comments and hope that we have answered each point to his/her satisfaction.

Thank you very much for your consideration of this manuscript for publication. 

 Yours Sincerely,

 Subhi J. Al’Aref, MD, FACC (Corresponding Author)

---

## [Decision Letter · Decision Letter 1]

15 Jul 2020

Extraction of radiographic findings from unstructured thoracoabdominal computed tomography reports using convolutional neural network based natural language processing

PONE-D-20-06619R1

Dear Dr. Al’Aref,

We’re pleased to inform you that your manuscript has been judged scientifically suitable for publication and will be formally accepted for publication once it meets all outstanding technical requirements.

Kind regards,

Yifan Peng, Ph.D.

Academic Editor

PLOS ONE

Additional Editor Comments (optional):

Congratulations! As suggested by one of the reviewers, please submit the high-resolution figures in the final submission.

Reviewers' comments:

Reviewer's Responses to Questions

**Comments to the Author**

1. If the authors have adequately addressed your comments raised in a previous round of review and you feel that this manuscript is now acceptable for publication, you may indicate that here to bypass the “Comments to the Author” section, enter your conflict of interest statement in the “Confidential to Editor” section, and submit your "Accept" recommendation.

Reviewer #1: All comments have been addressed

Reviewer #2: All comments have been addressed

2. Is the manuscript technically sound, and do the data support the conclusions?

Reviewer #1: Yes

Reviewer #2: Yes

3. Has the statistical analysis been performed appropriately and rigorously? 

Reviewer #1: Yes

Reviewer #2: Yes

4. Have the authors made all data underlying the findings in their manuscript fully available?

Reviewer #1: No

Reviewer #2: Yes

5. Is the manuscript presented in an intelligible fashion and written in standard English?

Reviewer #1: Yes

Reviewer #2: Yes

6. Review Comments to the Author

Reviewer #1: The authors addressed most of the comments well, except that I do not think the figures meet the publication requirement. I hereby recommend the editor to consider this manuscript for publication once the resolution issues are fixed.

Reviewer #2: The manuscript is greatly improved upon revision. I am satisfied with the authors' changes and responses to my comments.

7. PLOS authors have the option to publish the peer review history of their article (what does this mean?). If published, this will include your full peer review and any attached files.

Reviewer #1: No

Reviewer #2: No

---

## [Editor Report · Acceptance letter]

17 Jul 2020

PONE-D-20-06619R1 

Extraction of radiographic findings from unstructured thoracoabdominal computed tomography reports using convolutional neural network based natural language processing 

Dear Dr. Al’Aref:

I'm pleased to inform you that your manuscript has been deemed suitable for publication in PLOS ONE. Congratulations! Your manuscript is now with our production department. 

Kind regards, 

on behalf of

Dr. Yifan Peng 

Academic Editor

PLOS ONE